# RNA-Guided Genomic Localization of H2A.L.2 Histone Variant

**DOI:** 10.3390/cells9020474

**Published:** 2020-02-18

**Authors:** Naghmeh Hoghoughi, Sophie Barral, Sandrine Curtet, Florent Chuffart, Guillaume Charbonnier, Denis Puthier, Thierry Buchou, Sophie Rousseaux, Saadi Khochbin

**Affiliations:** 1CNRS UMR 5309; Inserm, U1209; Université Grenoble Alpes; Institute for Advanced Biosciences, F- 38700 Grenoble, France; naghmeh.hoghoughi@univ-grenoble-alpes.fr (N.H.); sophie.barral@univ-grenoble-alpes.fr (S.B.); sandrine.curtet@univ-grenoble-alpes.fr (S.C.); florent.chuffart@univ-grenoble-alpes.fr (F.C.); thierry.buchou@univ-grenoble-alpes.fr (T.B.); sophie.rousseaux@univ-grenoble-alpes.fr (S.R.); 2TGML, platform IbiSA; Aix Marseille Univ, Inserm U1090, TAGC, 13288 Marseille, France; guillaume.charbonnier@outlook.com (G.C.); denis.puthier@univ-amu.fr (D.P.)

**Keywords:** histone dynamics, testis-specific histone variants, epigenetic inheritance, repetitive elements, centromere

## Abstract

The molecular basis of residual histone retention after the nearly genome-wide histone-to-protamine replacement during late spermatogenesis is a critical and open question. Our previous investigations showed that in postmeiotic male germ cells, the genome-scale incorporation of histone variants TH2B-H2A.L.2 allows a controlled replacement of histones by protamines to occur. Here, we highlight the intrinsic ability of H2A.L.2 to specifically target the pericentric regions of the genome and discuss why pericentric heterochromatin is a privileged site of histone retention in mature spermatozoa. We observed that the intranuclear localization of H2A.L.2 is controlled by its ability to bind RNA, as well as by an interplay between its RNA-binding activity and its tropism for pericentric heterochromatin. We identify the H2A.L.2 RNA-binding domain and demonstrate that in somatic cells, the replacement of H2A.L.2 RNA-binding motif enhances and stabilizes its pericentric localization, while the forced expression of RNA increases its homogenous nuclear distribution. Based on these data, we propose that the specific accumulation of RNA on pericentric regions combined with H2A.L.2 tropism for these regions are responsible for stabilizing H2A.L.2 on these regions in mature spermatozoa. This situation would favor histone retention on pericentric heterochromatin.

## 1. Introduction

The transformation of nucleosomes, the universal basis of genome organization in eukaryotes, into a new protamine-based DNA packaging system during spermatogenesis remains a fascinating open question in biology. In mammals, a near genome-wide histone removal occurs during the postmeiotic stages of spermatogenesis in specific cell types known as elongating and condensing spermatids. Histone removal is also preceded by a genome-wide histone acetylation, especially on H4 lysines 5 and 8 [1], creating a specific binding site for the first bromodomain of the testis-specific BET factor, Brdt. We previously demonstrated that Brdt is, in turn, required for histone removal, and is involved in the loading of transition proteins and protamines into nucleosomes [1,2,3]. Our work on the involvement of histone variants in this process also led to the discovery of a series of testis-specific H2A and H2B histone variants as potentially interesting candidates in the control of postmeiotic male genome organization [4]. One of these variants, which we named H2A.L.2, displays the remarkable property of being expressed during late spermatogenesis at a time when most histones are removed [4]. H2A.L.2 therefore appeared as a particularly interesting candidate for further studies. Our previous investigations demonstrated that this histone H2A variant dimerizes with the testis-specific H2B variant, TH2B [4,5], and is coexpressed with transition proteins [6]. These studies questioned the dogma of the successive replacement of histones, stating that histones are first replaced by transition proteins, which are then themselves replaced by protamines. Indeed, our in vitro and in vivo data showed that H2A.L.2-TH2B incorporation opens the nucleosomes, allowing both transition proteins and protamines to be loaded on chromatin. These data suggest that transition proteins do not displace the full histone complement from DNA, but instead, buffer and control the action of protamines, which are the true histone displacers [6].

Although these investigations have provided important clarifications regarding the molecular basis of the transformation of the nucleosomes into nucleoprotamines, the issue of histone retention remained open. Many laboratories undertook the mapping of residual remaining histones on the genome of mature spermatozoa from different species, with sometimes contradictory conclusions [7,8,9,10,11,12,13]. After the identification of H2A.L.2 in our laboratory, microscopy approaches following immunofluorescence and FISH suggested that pericentric heterochromatin regions may represent privileged sites for histone retention. These observations were then supported by two independent publications based on high-resolution nucleosome mapping in mature spermatozoa [8,11], with further discussions on the validity of the bio-informatics presented in one of these publications [14].

Here, we demonstrate that H2A.L.2 nuclear localization is controlled by two properties of the protein: its intrinsic ability to target pericentric regions and its ability to bind RNA through its N-terminal arginine-rich motif. The late and localized accumulation of pericentric RNAs during spermatogenesis explains the previous observation of a preferential and stable association of H2A.L.2 at the pericentromeres and supports the hypothesis of enhanced histone retention in the pericentric regions of mature spermatozoa.

## 2. Materials and Methods

### 2.1. Spermatogenic Cell Fraction Purification

Spermatogenic Cell fraction purification was performed as previously described in [6].

### 2.2. Quantification of Expression of Major Satellite Sequences in Male Germ Cells

#### 2.2.1. RNA Extraction from Male Germ Cells

Spermatocytes, round and condensing spermatids, were purified on a BSA gradient as previously described [6]. RNA was extracted using Trizol (Life Technologies, Carlsbad, CA, USA) reagent. Cells were centrifuged for 5 min at 1500 rpm at 4 °C, 750 μL of Trizol was added to the pellet, and cells were resuspended. The suspension was passed through a G26 needle 10 times and incubated for 5 min at room temperature. After 5 s centrifugation (to settle the foam), 200 μL of chloroform was added to the samples, shaken gently by hand for 15 s, and incubated for 15 min at room temperature. Samples were centrifuged at 12,000 rpm for 15 min at 4 °C and the aqueous (upper) phase was collected. Then, 500 μL of chloroform was added to the samples, which were subsequently incubated for 3 min at room temperature and centrifuged at 12,000 rpm at 4 °C for 15 min. Next, 10 μL of glycogen was added to the aqueous phase and tubes were inverted 4 times. Then, 500 μL of Isopropanol (500 μL/1 mL Trizol) was added to the samples, and the tubes were gently inverted four times, incubated for 10 min at room temperature, and centrifuged at 12,000 rpm for 15 min at 4 °C. The supernatant was discarded, 1 mL of 75% ethanol was added to the samples, and a vortex step enabled detaching the pellets, which were centrifuged at 7500 rpm at 4 °C for 5 min. The pellets were dried out and resuspended in 20 μL of (RNase free, MN, Düren, Germany) H_2_O and the RNA concentration were measured by Thermo Scientific™ Spectrophotometer NanoDrop™ 2000/2000c (Thermofisher, Waltham, MA, USA).

#### 2.2.2. Sequencing

RNA was extracted from the spermatocytes, round and condensing spermatids, of H2A.L.2-WT and H2A.L.2-KO mice. For the three cell types, the respective total amounts obtained were 30 μg, 52.41 μg, and 16 μg for WT mice, and 22.96 μg, 46.28 μg, and 16.3 μg for KO mice.

RNA for Removal of most rRNA was proceeded using the NEBNext rRNA Depletion Kit Ribodepletion. Quality was verified by Agilent Bioanalyzer 2100. Libraries were prepared using the NEBNext Ultra Directional RNA Library Prep kit for Illumina, and stranded paired-end 2 × 150 bp reads were produced by Illumina NextSeq500 (Illumina, San Diego, CA, USA). Base calling was performed using RTA version 2 (https://support.illumina.com).

#### 2.2.3. Post Processing of Fastq Files

Paired fastq files were trimmed using fastx_trimmer (http://hannonlab.cshl.edu/fastx_toolkit/) (with options -l 30 -Q33, meaning 5 prime trimming keeping 30 bp-length fragments). The trimmed fastq files were aligned on the USCS mm10 genome using bowtie2 (with the following options: –end-to-end, –no-mixed, –no-discordant). The UCSC Table browser Repeatmasker track was downloaded from http://genome.ucsc.edu for mm10 assembly of mouse genome in a BED format and used to select genome regions with GSAT_MM annotation, which resulted in 84 features. These features were exported as a SAF file named gsat_mm_feat.saf. The reads associated with these features were counted using featureCounts (with options -a gsat_mm_feat.saf -F SAF –largestOverlap -Q 0 -T 8 -o gsat_mm_feat_rna_count_0.txt). Sample read counts were normalized into reads per million of total reads per sample, and the pseudo-log of this RPM normalized value were plotted in a box-and-whisker fashion using R.

### 2.3. Cell Culture and Transfection

Firstly, 1.5 × 10^5^ NIH 3T3 (Mouse embryonic fibroblasts) cells were cultured in two-well glass cover lab-tek containers in DMEM 1g/L D-glucose supplemented with 10% newborn calf serum NBCS + 2% L-glutamine and 1% penicillin/streptomycin. Cells were maintained at 37 °C in a humidified incubator at 5% CO2. Cells were transfected with 1μg/μL of each of the plasmids mentioned in Table 1, for 24 h using lipofectamine 2000 (Invitrogen, Carlsbad, CA, USA) and optimum (Gibco, Life Technologies, Carlsbad, CA, USA).

### 2.4. RNA Fluorescence In Situ Hybridization

#### 2.4.1. Preparation of Probes

For RNA FISH (Table 2), 0.05mM of FITC-labelled/Cy-3-labelled forward strand and FITC-labelled/Cy3-labelled reverse strand probes (EXIQON) were used. The probes were precipitated in 1 μg of salmon sperm DNA (Invitrogen), 5 μL of sodium-acetate (NaAc), and 100 μL of ethanol (100%) per 150 μL reaction. The precipitation mix was kept at −80 °C for 30 min. Samples were centrifuged for 20 min at 12,000 rpm at 4 °C. The precipitated DNA was washed twice in 70% ethanol and then air-dried. For each probe, the pellet was resuspended thoroughly in 10 μL of hybridization buffer (20% formamide, 10% dextran sulfate, SSC2X). The probe was denatured for 5 min at 75 °C and kept on ice during cell preparation.

#### 2.4.2. Hybridization

Twenty-four hours posttransfection, cells were rinsed once with RNase-free PBS 1X. Next, cells were permeabilized in freshly made 0.5% Saponine/0.5% Triton (SIGMA) buffer for 5 min on ice and were fixed in freshly made 4% formalin solution (SIGMA, ST. LOUIS, MO, USA) for 10 min at room temperature. Three washes in RNase-free PBS1X were carried out at room temperature. Prior to FISH, cells were rinsed twice in ethanol 70% and dehydrated in 90% and 100% ethanol for 5 min each at room temperature. Cells were then air-dried; denatured probes were deposited on the plastic slides and covered with 22 × 22mm RNase-free glass coverslips. Slides were incubated in a humidified chamber at 37 °C. After 24 h incubation, coverslips were gently removed. Three washes were carried out using formamide 15% / 2X SSC and 1XSSC, respectively, for 5 min each at room temperature. Cells were washed once in PBS1X for 5 min at room temperature. DNA was stained using DAPI (1/500), and slides were mounted using Dako, as previously described.

### 2.5. Immunofluorescence on Germ Cells

Mature epididymal sperm cells were obtained by swim-up, as previously described in [9], and dried out on glass slides. Immunofluorescence using rabbit polyclonal anti-H2A.L.2 [6] and mouse monoclonal antiprotamine P2 (Briar Patch Biosciences (cat HUP2B) were performed as described previously [6].

### 2.6. Microscope Analysis and Image Processing

We acquired brightfield and fluorescent images of spermatogenic and NIH 3T3 cells under a Zeiss inverted microscope, objective 63× (Carl Zeiss, Oberkochen, Germany). We used Adobe Photoshop CS3 and ImageJ for further processing.

### 2.7. Ethics

For this study, the animals were bred in an animal facility with an official and up-to-date agreement according to French and European regulation, and euthanized following a procedure approved by ad hoc committees of the Grenoble Alpes University. All investigators have official animal-handling authorization, obtained after two weeks of intensive training and a final formal evaluation. Efforts were made to minimize suffering.

## 3. Results

### 3.1. H2A.L.2 Has the Intrinsic Property to Target Pericentric Heterochromatin

We previously showed that H2A.L.2 is expressed at the same time as transition proteins, and is required for the loading of transition proteins onto nucleosomes [6]. However, considering the spermatogenic cell stages that follow histone removal, a fraction of H2A.L.2 survives in restricted regions of the nucleus in condensing spermatids, as well as in epididymal spermatozoa (Figure 1a and b). Careful consideration of the subnuclear domains associated with the surviving H2A.L.2 reveals that they largely correspond to the DAPI-bright regions that are known to be mostly enriched in pericentric major satellite DNA (Figure 1a). This situation persists in mature spermatozoa (Figure 1b). However, the codetection of H2A.L.2 and protamine 2 using confocal microscopy indicated that the pericentric regions do not exclusively contain these histone-based structures, but that they are also largely protaminized (Figure 1b).

In order to better understand the molecular basis of this region-specific accumulation of H2A.L.2 during late spermatogenesis, we decided to characterize the molecular properties of H2A.L.2 in terms of its preferential nuclear localization.

For this purpose, we used somatic cells, a totally irrelevant system, to investigate whether the pericentric localization of H2A.L.2 was specific to the context of late spermatogenic cells, or if it could also be observed in an unrelated context upon its ectopic expression in somatic cells.

To this end, we chose NIH3T3 cells, which display clear DAPI-bright foci (chromocenters), to ectopically express H2A.L.2. In parallel, we expressed H2A.B.3, a short H2A histone variant, which is structurally close to H2A.L.2, but, in contrast to H2A.L.2, is expressed in meiotic and early postmeiotic cells, and disappears later when H2A.L.2 is expressed [15]. We also used the replication-dependent H2A as another control.

Figure 1c shows that H2A.L.2 is mostly observed in the pericentric regions when expressed out-of-context in somatic cells. Under the same conditions, despite important sequence and structural similarities between H2A.L.2 and H2A.B.3, the latter was unable to concentrate in the pericentric regions, and showed a similar pattern of nuclear distribution as H2A.

Based on this experiment, we conclude that H2A.L.2 has the intrinsic property to target the pericentric heterochromatin, and that no other testis-specific factor is required for this particular genomic targeting of H2A.L.2.

An additional control consisted in the expression of a chimeric protein bearing a major satellite-specific zinc finger fused to the VP16 activation domain (pTVLp64) [16]. The expression of this protein leads to a complete disorganization of the chromocenters, as can be judged by the dispersion of the DAPI-bright regions in the nucleus. Interestingly, under these conditions, the ectopically expressed H2A.L.2 could not form any nuclear domain (Figure 1c, asterix), indicating that the observed H2A.L.2 domains are not protein aggregates, but are dependent on structured chromocenters. It is of note that a close inspection of these cell nuclei shows that H2A.L.2 still follows the dispersed DAPI-bright DNA regions (Figure 1c, pTVLp64 panel, compare RFP and DAPI panels).

### 3.2. Pericentric Localization of H2A.L.2 is Controlled by its N-terminal RNA-Binding Motif

H2A.L.2 belongs to a group of short H2A variants that all lack the typical H2A acidic patch, and present a shorter C-terminal docking domain than the canonical H2As [4,15,17,18]. Additionally, H2A.L.2 shares with H2A.B.3 a remarkable enrichment in arginine residues in their N-terminal region (Figure 2a), which in H2A.B.3 was shown to be involved in binding RNA [19].

In order to test the role of this putative RNA-binding domain at the N-terminal of H2A.L.2, we replaced this region by the corresponding region of H2A (scheme, Figure 2a).

The ectopic expression of this mutant of H2A.L.2 (H2A.L.2-H2Anter) in NIH3T3 cells unexpectedly showed an increase in cell number with chromocenter localization of the protein. Indeed, compared to wild-type H2A.L.2, the localization of the protein in the nucleoli, as well as the nuclear genomic background levels of H2A.L.2, significantly decreased, leading to an enhanced chromocenter localization of this H2A.L.2 mutant (Figure 2a). A decrease in the accumulation of the mutant H2A.L.2 in nucleoli is itself a sign of a decreased nonspecific RNA-binding activity compared to that of wild-type H2A.L.2. Indeed, it is known that the high concentration of RNAs in nucleoli attracts various RNA-binding proteins [20].

These data indicate that the N-ter RNA-binding domain of H2A.L.2 could be an important element in controlling the nuclear localization of the protein. In our test model of NIH3T3 cells, nonspecific RNA-binding by H2A.L.2 could decrease the tropism of H2A.L.2 for the transcriptionally silent RNA-poor heterochromatin regions, probably by mediating the stabilization of H2A.L.2 on other genomic regions via chromatin-bound RNAs [21].

### 3.3. The Dynamic Turnover of H2A.L.2 is Controlled by its N-terminal RNA Binding Domain

To better characterize the role of this N-terminal domain in the nuclear localization of H2A.L.2, we also set up a Fluorescent Recovery After Photo-bleaching (FRAP) approach to measure and compare the stability of the association of the wild-type and mutant form of H2A.L.2 with the pericentric regions. To this end, we first established NIH3T3 cell lines stably expressing wild-type RFP-H2A.L.2 or a mutant form of the protein bearing a replacement of its N-terminal region by that of H2A (H2A.L.2-H2Anter). After photo-bleaching, the recovery of wild-type H2A.L.2 at the pericentric regions was much faster than that of the mutant protein (Figure 2b). This result indicates that H2A.L.2 without its RNA-binding domain remains more stably bound to the pericentric heterochromatin than the wild-type H2A.L.2.

These data nicely support our conclusions on the role of H2A.L.2 RNA-binding domain in the intranuclear dynamics of H2A.L.2. Indeed, as shown in Figure 2a, the RNA-binding domain of H2A.L.2 keeps a fraction of the protein in the extra-chromocenter regions of the genome. This pool of H2A.L.2 can therefore guarantee a supply for this histone variant after photobleaching, and therefore, a dynamic exchange of the protein between the pericentric heterochromatin and the rest of the genome. This dynamics of H2A.L.2 localization is reduced upon the replacement of H2A.L.2 N-terminal by that of H2A because of its inability to be retained on nonpericentric chromatin and to constitute a pool of mobile nuclear H2A.L.2 (scheme, Figure 2c).

A prediction of this model is that the ectopic expression of RNA in our test NIH3T3 cells stably expressing wild-type or the non-RNA-binding mutant of H2A.L.2 should result in the absence of protein accumulation in the chromocenters of wild-type H2A.L.2, but should not affect the localization of the mutant H2A.L.2.

In order to test this hypothesis, two repeat units of the mouse pericentric DNA were cloned in an expression vector and transfected in NIH3T3 cells stably expressing either wild-type or the non-RNA-binding mutant of H2A.L.2. Figure 2d shows that, as predicted, the ectopic RNA expression in these cells reduces the accumulation of wild-type H2A.L.2 in chromocenters, while this forced RNA expression had no effect on the localization of the non-RNA-binding mutant of H2A.L.2.

### 3.4. Enhanced Major Satellite RNA Expression and Localization during Late Spermatogenesis

Our investigation of H2A.L.2 nuclear localization in NIH3T3 cells, a context totally irrelevant to spermatogenesis, showed that the dynamics and efficiency of H2A.L.2 localization at the chromocenter regions depends on the opposed properties of this H2A variant, i.e., its RNA-binding activity and its intrinsic ability to target pericentric heterochromatin. However, in the context of the final stages of spermatogenesis, these two properties of H2A.L.2 could cooperate to stabilize the association of H2A.L.2 with pericentric regions. Indeed, an enhanced expression of major satellite RNA and its localized association with these regions could stabilize H2A.L.2 at the pericentric regions during late spermatogenesis.

We used an RNA-seq approach to compare the transcriptomes between spermatogenic cells at various stages of differentiation including spermatocytes, round spermatids, and condensing spermatids. To test the role of H2A.L.2 itself in inducing the expression of the major satellite RNA, we also considered the transcriptome of the corresponding H2A.L.2 KO cells. Figure 3a shows that in both wild-type and H2A.L.2 KO cells, the expression of satellite RNA significantly increased in condensing spermatids. These results showed that H2A.L.2 incorporation at the pericentric region is not needed for the activation of satellite RNA expression, since the increase in major satellite RNAs was also observed in the absence of H2A.L.2.

Accordingly, in NIH3T3 cells, we also observed that the pericentric localization of H2A.L.2 did not induce the expression of these regions, while the expression of pTVLp64, capable of targeting pericentric regions, efficiently induced pericentric RNA expression, as can be judged by the accumulation of both forward and reverse major satellite transcripts (Figure 3b).

An increased targeted accumulation of satellite RNAs at the pericentric regions could therefore reinforce the intrinsic ability of H2A.L.2 to associate with these regions.

Accordingly, we set up a FISH-based detection of major satellite RNA on the H2A.L.2-WT epididymal maturing spermatozoa. Figure 3c shows a localized accumulation of sense and antisense major satellite RNAs.

Considering all the data on the major satellite RNA expression and pericentric H2A.L.2 localization, it appears that when RNA accumulates at the pericentric regions, both the tropism of H2A.L.2 for pericentric regions and its RNA-binding activity cooperate to target H2A.L.2 to the pericentromeres and to stabilize its localization in these regions (Figure 3d).

## 4. Discussion

During the postmeiotic reorganization of the male genome, the invasion of H2A.L.2-containing nucleosomes by protamines, modulated by a simultaneous incorporation of transition proteins, leads to the generation of transitional genome organizational states [6]. In agreement with other published data on the short H2A variants [15,18,22,23,24,25], H2A.L.2 generates open nucleosomes protecting only about 130 bp of DNA against MNase digestion [6]. Similar to CENP-A containing nucleosomes, H2A.L.2-containing open nucleosomes create a platform for the loading of nonhistone proteins onto nucleosomes. We previously reported that in the case of H2A.L.2 nucleosomes, these nonhistone proteins are transition proteins and protamines, with transition proteins not directly replacing histones, but buffering and controlling histone replacement by protamines [6]. Following its genome-wide action, H2A.L.2 continues to be detected in the DAPI-dense pericentric regions of the genome in condensing spermatids and mature spermatozoa ([4]; this work).

This finding has important implications on the controversial data published by different groups regarding the specific positions of retained nucleosomes in mature spermatozoa [7,8,9,10,11,12,13]. Indeed, the preferential genomic position of retained nucleosomes in sperm cells is still a subject of debate [26]. Our work supports the conclusions formulated by two of these groups [8,11], on the preferential association of histones with the repetitive elements, especially the major satellite regions. Indeed, our low-resolution microscopy investigations revealed that pericentric regions display a specific organization characterized by the preferential retention of H2A.L.2, which persists from condensing spermatids to the final stages of maturing spermatozoa. Due to the repetitive nature of the pericentric genomic regions, it would be challenging to quantitatively estimate the amounts of H2A.L.2 present on these regions compared to the rest of the genome by using ChIP sequencing data; therefore, our conclusions are mostly based on in situ data. It is of note that the specific structural organization of the pericentric regions and various parameters such as antibody accessibility could also enhance this preferential detection of H2A.L.2-related immunofluorescence signal. Examples of such situations have been previously reported. For instance, it has been shown that the almost exclusive enrichment of macroH2A histone variant on the inactive X chromosome observed by microscopy was exaggeratedly enhanced compared to macroH2A enrichment on the X chromosome, as measured by ChIP-seq [27,28]. Therefore, although H2A.L.2 is preferentially detected over the pericentric regions in situ, we also expect its presence on other genomic regions, in agreement with the detection of nucleosomes on genomic regions other than pericentric heterochromatin observed by all investigators.

Despite this cautionary note, the preferential accumulation of H2A.L.2 in the pericentric regions is strongly supported by the remarkable property of H2A.L.2, which is capable of targeting pericentric regions upon its ectopic expression in the completely unrelated context of somatic cells. In this respect, the case of H2A.L.2 is not unique, since an *Arabidopisis* H2A histone variant, H2A.W, has also been shown to target heterochromatin regions [29]. However, in the case of H2A.W, as well as for H2A.L.2, the mechanisms underlying this specific targeting remain to be understood. It should be noted that, in contrast to H2A.L.2, H2A.W does not seem to bear any RNA-binding domain and does not show any RNA-dependent incorporation and dynamics. H2A.L.2 presents the capacity, not shared with H2A.W, to modulate its ability to target the heterochromatic regions thanks to its RNA-binding domain. Indeed, inspired by a published work on another member of the short H2A variants expressed in spermatocytes and in round spermatids, H2A.B.3 [19], we demonstrated here that H2A.L.2 targeting of the pericentric heterochromatin can be controlled by RNA binding. This conclusion is supported by the fact that in somatic cells, the replacement of H2A.L.2 putative RNA-binding domain enhances its pericentric localization, and hence, reduces its dynamics.

In late spermatids and in spermatozoa, as opposed to NIH3T3 cells, the pericentric regions not only become expressed, but also the corresponding RNAs remain associated with these regions. Based on our detailed analysis of H2A.L.2 nuclear localization in somatic cells, we propose that in sperm cells, both the tropism of H2A.L.2 for the pericentric regions and its RNA binding activity cooperate to stabilize this variant in the pericentric regions.

Additionally, the co-immunodetection of protamine 2 and H2A.L.2 shows that, despite pericentric regions being preferentially associated with H2A.L.2, these regions are also protaminized, suggesting that, even within these regions, a mixture of different types of DNA-packaging structures should be present. This particular organization of the pericentric regions involving H2A.L.2 could have important implications for the structuration and transcription of the male heterochromatin postfertilization [30,31], but this question, similar to the issue of position-specific nucleosome retention discussed by many groups, remains open.

## Figures and Tables

**Figure 1 cells-09-00474-f001:**
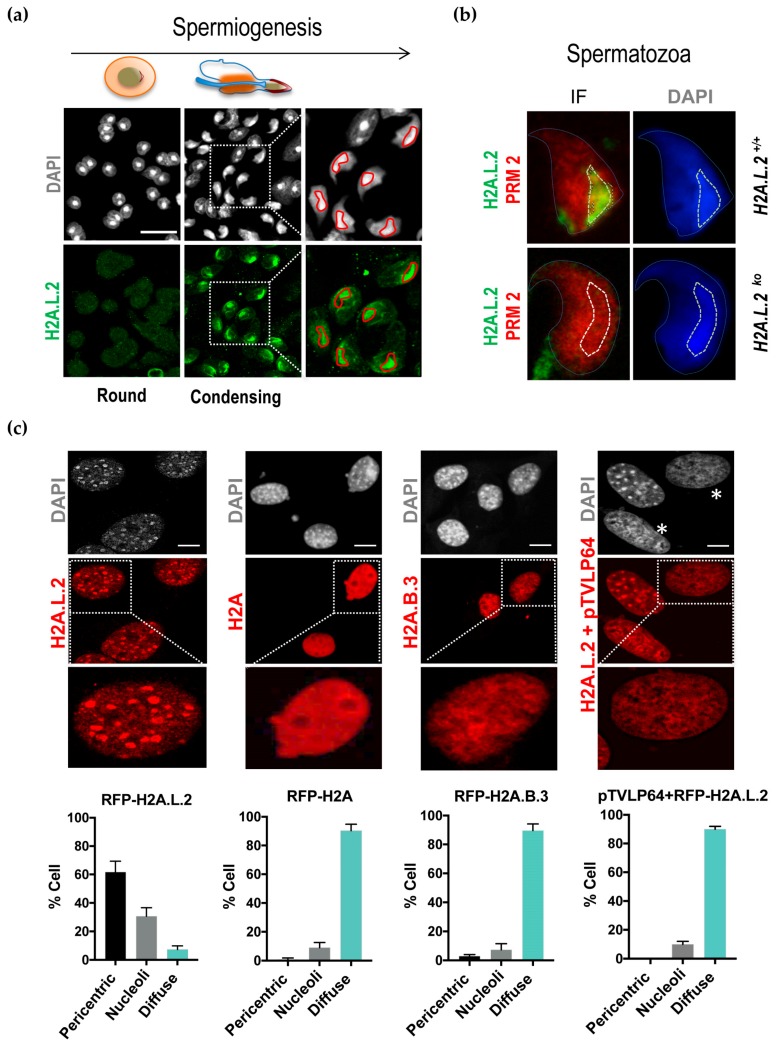
Pericentric localization of H2A.L.2 in condensing spermatids. The intranuclear localization of H2A.L.2 was monitored by immunofluorescence during successive stages of postmeiotic spermatogenesis (**a**) and in mature spermatozoa from H2A.L.2-WT and H2A.L.2-KO mice (**b**) [6]. (**a**) Round and condensing spermatids after DAPI-staining (upper panels) and H2A.L.2 immunostaining (lower panels) are shown. The indicated squares are magnified and shown on the right. The DAPI-bright regions are delimited by a red line. Scale bars represent 10 μm. (**b**) Co-immunodetection of H2A.L.2 (green) and protamine 2 (red) in the epididymal spermatozoa is shown. The dotted lines delimit the pericentric regions. (**c**) The indicated RFP-tagged H2As were expressed in NIH3T3 cells and the intranuclear localization of the proteins visualized. In one experiment pTVLP64 synthetic protein corresponding to the major satellite-specific zinc finger fused to the VP16 activation domain was coexpressed with RFP-H2A.L.2. The histograms below represent the proportions of cells with the indicated intranuclear localization of the considered histones. The *y*-axis values represent the proportions of cells with a diffuse signal (“Diffuse”) or a signal localized in DAPI-bright regions (“pericentric”) or in the nucleoli (“Nucleolei”), obtained from the analysis of three independent experiments for each condition (200 cells were counted per experiment). The latter (nucleoli) localization corresponds to nuclear regions accumulating RFP-H2A.L.2 in the relatively large domains (about 2 or 3 per nucleus), other than the DAPI-bright chromocenters. Error bars represent standard deviations. When compared to RFP-H2A.L.2 experiment, the decreases in cell numbers with pericentric localization and increases in cell numbers with diffuse localization were all significant with t-test *p*-values < 0.01, whereas decreases of nucleoli localization were of borderline significance in RFP-H2A, RFP-H2A.B.3 and pTVLP64 + H2A.L.2 expressing cells with respective *p*-values of 0.05; 0.02 and 0.04. Scale bars represent 10 μm.

**Figure 2 cells-09-00474-f002:**
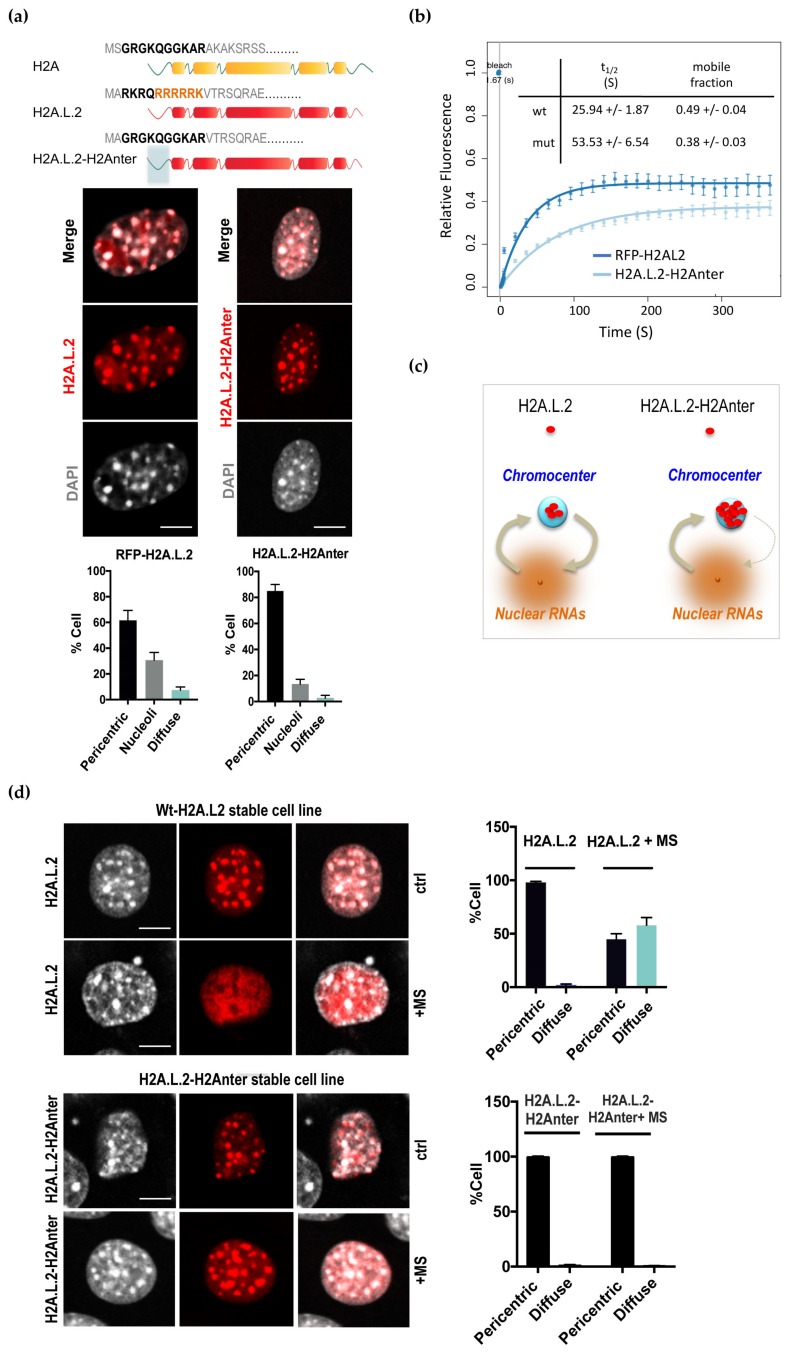
RNA-controlled dynamic pericentric localization of H2A.L.2. (**a**) NIH3T3 cell transiently expressing RFP-H2A.L.2 or a mutated version of RFP-H2A.L.2 with the replacement of its N-terminal sequence with that of H2A, named H2A.L.2-H2Anter, were obtained and the intracellular localization of the expressed proteins was monitored. The scheme in the upper panel represents the N-terminal regions of H2A.L.2 and the amino-acid sequence replaced in the H2A.L.2-H2Anter is indicated in bold characters. The R and K amino-acids in H2A.L.2 are also highlighted in red. The histograms represent the proportions of cells showing the indicated intranuclear localizations of the considered proteins. The Y-axis values represent the proportions of cells with a signal localized in DAPI-dense regions (pericentric) or in the nucleoli, or showing a diffuse localization, obtained from the analysis of three independent experiments for each condition (200 cells were counted per experiment) as in Figure 1c. Compared to H2AL2-RFP, H2AL2-H2Anter expressing cells show an increase in the proportion of cells with pericentric localization (t-test *p*-value = 0.005) and decrease in the proportions of cells with nucleoli localization (*p* = 0.018), whereas no significant difference was observed in the proportions of cells with diffuse localization between the RFP-H2A.L.2 and H2A.L.2-H2Anter expressing cells (*p*-value = 0.11) (**b**) NIH3T3 cell lines stably expressing RFP-H2A.L.2 or H2A.L.2-H2Anter, were established and were used to monitor protein dynamics by Fluorescence Recovery After Photo-bleaching (FRAP). Ten RFP-labelled chromocenter foci (wild-type H2A.L.2 or mutated) were photo-bleached and the recovery of fluorescence monitored. The ten datasets of each experiment were individually fitted and used to calculate the half-life (t1/2) of fluorescence recovery and the mobile fractions (indicated in the inset). The mean and standard deviation measured at successive time points from the ten datasets are plotted. (**c**) The scheme depicts the working hypothesis on the role of the H2A.L.2 putative RNA-binding domain in the dynamics of H2A.L.2 as measured by FRAP. Nuclear RNAs would ensure the existence of a pool of RNA-bound H2A.L.2 out of the chromocenters, which would supply the chromocenters with H2A.L.2 after photo-bleaching due to the intrinsic capacity of H2A.L.2 to target the pericentric heterochromatin. H2A.L.2-H2Anter, unable to interact with RNA, would be directly targeted to the pericentric regions leaving the cells devoid of any pool of immediately accessible H2A.L.2 to repopulate the chromocenters after photo-bleaching. (**d**) A fragment of major satellite DNA was cloned in an expression vector under the CMV promoter (named “MS”) and expressed in the aforementioned cells lines stably expressing H2A.L.2 wild type and the mutant form. The left panels show the intracellular localization of the expressed proteins monitored by immunofluorescence. Scale bars represent 10 μm. The bar plots on the right panel show the respective proportions of nuclei corresponding to each of the two categories of intranuclear distributions of signal for H2A.L.2 (upper bar plots) and H2A.L.2-H2Anter (lower bar plots) and the values were calculated as explained above. The proportions of H2AL2 expressing cells with pericentric or diffuse localization as a result of major satellite RNA expression, were significantly different (t-test *p*-values < 0.01), which was not the case for H2AL2-H2Anter expressing cells, where the signal remained predominantly “pericentric”.

**Figure 3 cells-09-00474-f003:**
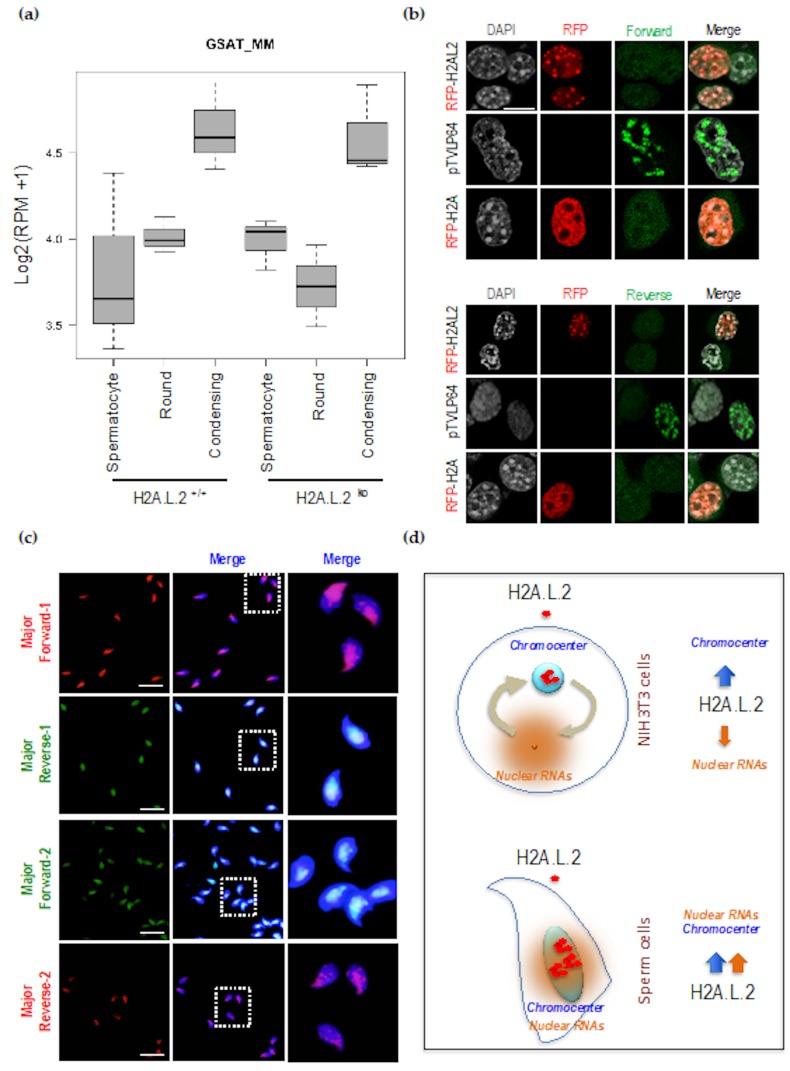
H2A.L.2 pericentric chromatin targeting does not induce RNA expression, but the localized expression of pericentric RNA is associated with the pericentric targeting of H2A.L.2 in spermatozoa. (**a**) Total RNAs from the indicated fractionated spermatogenic cells, spermatocytes, round and condensing spermatids, from H2A.L.2-wild type (+/+) or H2A.L.2-KO mice were sequenced and aligned. Pseudo-log of normalized read counts per million reads corresponding to major satellite sequences were plotted in a box-and-whisker fashion using R. Each box plot represents three samples of cells isolated from three different experiments. The expression of major satellite RNA was significantly increased in condensing spermatids compared to spermatocytes and round spermatids in both H2A.L.2-WT and H2A.L.2-KO mice (t-test *p*-values < 0.05). (**b**) NIH3T3 cells were transfected with RFP-H2A.L.2 or RFP-H2A expressing vectors, or by pTVLP64, and the expression of major satellite RNA was visualized by FISH with a forward and a reverse fluorescently-labeled probes (corresponding to the two first probes in the Table 2), detecting major satellite RNAs with both sense (upper panel) and antisense (lower panel) orientations. Scale bars represent 10 μm. (**c**) Four fluorescently-labelled probes (Table 2) were used to detect the expression of major satellite RNA in epididymal spermatozoa. The right panels show magnifications of the indicated fields. Scale bars represent 10 μm. (**d**) Schematic representation of our working hypothesis regarding the respective roles of RNAs in the control of H2A.L.2 intranuclear localization in somatic cells and in spermatozoa. In NIH3T3 cells, the presence of nuclear RNA outside the chromocenters results in a moderate pericentric accumulation of H2A.L.2. In condensing spermatids and in spermatozoa, the localized accumulation of major satellite RNA in pericentric regions cooperates with the intrinsic property of H2A.L.2 to target pericentric heterochromatin to stably maintain H2A.L.2 at the pericentric regions.

**Table 1 cells-09-00474-t001:** Recombinant DNA.

Plasmid	Source
pTag-RFP-H2A.L.2	This paper
pTag-RFP-H2A	This paper
pTag-RFP-H2A.L.2-H2Antere	This paper
pTag-RFP-H2A.B.3	This paper
pTVLP64-ms-Ca B15	Gift from Maria Elena Torres-Padilla
pLKO.1 major satellite tandem repeats	This Paper

**Table 2 cells-09-00474-t002:** RNA FISH oligonucleotides.

Oligonucleotide	Sequence (5′ to 3′)
Forward 1	FITC-TCTTGCCATATTCCACGTCC
Forward 2	Cy3-GATTTCGTCATTTTTCAAGT
Reverse 1	Cy3-GCGAGGAAAACTGAAAAAGG
Reverse 2	FITC-GCGAGAAAACTGAAAATCAC

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
