# Peer review of "RNA-Guided Genomic Localization of H2A.L.2 Histone Variant"

_cells, 2020, doi:10.3390/cells9020474_

Round 1
Reviewer 1 Report
The manuscript describes the subcellular localization of the histone variant H2A.L.2. It is demonstrated that H2A.L.2 specifically localizes at pericentric regions. It is further demonstrated that H2A.L.2 can bind RNA via its N-terminal arginine-rich motif. It is speculated that accumulation of RNA at pericentric regions stabilizes H2A.L.2 on these regions, important for histone retention.
The manuscript is well written. The setup of the figures is however chaotic, with labeled panels (a,b, c etc) that are themselves composite figures without much explanation. Furthermore, the data presented are not always convincing and open to different interpretations than those made by the authors.
Major comments:
Several abbreviations used are not needed and only make the manuscript less appealing to read. My suggestion would be not to abbreviate transition proteins (in the manuscript abbreviated to HP) or protamines (manuscript abbreviated as PRMs). I do not see the added value of these abbreviations.
For clarity it seems better as more panels are differently labelled in the figures. A sit stand now it is chaotic
Of the 34 papers that this manuscript refers to, 11 are self-citations. Although with a specific subject as histone variant self-citations can usually not be avoided, >30% of self-citations is not realistic. I think citations 1-6 can be removed (and the text referring to these citations with losing the message of the manuscript.
There is a lack of consistency in the Materials and Methods section. Minutes are abbreviated to min, but seconds are not abbreviated. Hours are abbreviated to h (line 111, or not abbreviated (line128).Capitals are used when not needed (Glycogen line 81, Calf line 108, Streptomycin line 109, Ethanol line 117) and at places ethanol is used, while at other places ETOH is used (suggested to only use ethanol). Both μl and μL are used. Both are fine with as long as it is consistent.
In the description of probe preparation it is stated that forward strands were FITC-labelled and reverse strand Cy-3 labelled. Table 2 suggest that forward is either FITC or Cy3, same for remove . please correct. In table 2, size of the product (in bp!?) is not relevant and should be removed. In the text it can simply be mentioned that 20 nt probes were used.
Line 138 …as previously described, please add reference.
Figure 1.
a) The legend states that the pericentric regions are indicated by a red line (left) or dotted lines (right). Probably, intense DAPI-stained areas are considered pericentric regions, but this is subjective and it should be stated as such (so DAPI-rich instead of pericentric regions). The M&M mentions Hoechst instead of DAPI.
b) The words on the x-bars are barely readable, please correct. Also, indicate which bars are significantly different.
Figure 2
a)The AA sequence of H2A.L.2-H2Anter that replaces the N-terminal part of H2AL.2 should be presented, next to the sequence that is replaced.
b-c) Words on x-axes cannot be read, please change. Also indicate levels of significance next to the error bars.
b) It is stated that the dynamics of H2A.L.2 is lost upon the replacement of the N-terminal part with that of H2A (line 265). Based on Figure 2b it is clear that it is not lost, but reduced.
c) The pictures and the graphs do not seem to match. There seems to be quite a bit of diffuse staining in the H2A.L.2-H2aAnter cells, but I the graphs it is zero. Please explain the discrepancy. Also, the alignment of the pictures is sloppy, same for 3c.
Line 271 Indicate more specifically which satellite DNA was cloned into the expression vector.
Figure 3a Levels of significance should be indicated.
For the collection of sperm, mice were used. It should be confirmed that all experiments have been
Performed in accordance with relevant regulations, and mentioning the committee approving the experiments.
Minor comments
Articles are lacking at places: line 76 a G26 needle; line 77 at instead of in; line 90 the Qiagen; line 92 the NEB; line 118 for instead of by; line 120 the pellet.
Author Response
Comments and Suggestions for Authors (referee N°1)
Major comments:
Several abbreviations used are not needed and only make the manuscript less appealing to read. My suggestion would be not to abbreviate transition proteins (in the manuscript abbreviated to HP) or protamines (manuscript abbreviated as PRMs). I do not see the added value of these abbreviations.
Reply
The abbreviations TPs and PRMs have been replaced by “transition proteins” and “protamines”
For clarity it seems better as more panels are differently labelled in the figures. As it stands now it is chaotic
Reply
Most of the ex-subpanels are now renamed independently.
Of the 34 papers that this manuscript refers to, 11 are self-citations. Although with a specific subject as histone variant self-citations can usually not be avoided, >30% of self-citations is not realistic. I think citations 1-6 can be removed (and the text referring to these citations with losing the message of the manuscript.
Reply
We understand that too many self-citations should be avoided in general. However, it is also important to describe the context of the study, which is post-meiotic genome reorganization. Regarding this biological context, thefact that most of the papers focussing on this issue are from our laboratory increases the rate of self-citations. However, to comply with this referee’s request we deleted references 2-4. The references 1 and 6 cannot be removed since the reported data are important to understand the context of this work.
There is a lack of consistency in the Materials and Methods section. Minutes are abbreviated to min, but seconds are not abbreviated. Hours are abbreviated to h (line 111, or not abbreviated (line128). Capitals are used when not needed (Glycogen line 81, Calf line 108, Streptomycin line 109, Ethanol line 117) and at places ethanol is used, while at other places ETOH is used (suggested to only use ethanol). Both μl and μL are used. Both are fine with as long as it is consistent.
Reply
In the revised version we paid a particular attention to homogenise the text and make it consistent.
In the description of probe preparation it is stated that forward strands were FITC-labelled and reverse strand Cy-3 labelled. Table 2 suggest that forward is either FITC or Cy3, same for remove.
Reply
We are sorry for this confusion. Both probes were originally synthesized as indicated with both FITC and Cy3 but for the experiment shown in Fig. 3b, only two of them, both FITC - labelled, were used whereas in the experiments shown in Fig. 3C, all four probes were used.
This is now better explained in the text.
In table 2, size of the product (in bp!?) is not relevant and should be removed. In the text it can simply be mentioned that 20 nt probes were used.
Reply
This column was removed from the table
Line 138 …as previously described, please add reference.
Reply
Sorry for this misleading wording. “previously described” actually meant “described above”. It has now been removed.
Figure1. a) The legend states that the pericentric regions are indicated by a red line (left) or dotted lines (right). Probably, intense DAPI-stained areas are considered pericentric regions, but this is subjective and it should be stated as such (so DAPI-rich instead of pericentric regions).
Reply
We changed “pericentric” by “DAPI-bright”.
The M&M mentions Hoechst instead of DAPI.
Reply
Hoechst has been changed to DAPI
b) The words on the x-bars are barely readable, please correct. Also, indicate which bars are significantly different.
Reply
This could be due to alterations of our manuscript, which was not converted into pdf format, which could have caused the disorganizations seen by the referees.
The original, high-resolutionfigures not embedded in the text have now been submitted, which should circumvent this issue.
p-values representing the levels of significance have now been added in the legends for all histograms.
Figure2 a)The AA sequence of H2A.L.2-H2Anter that replaces the N-terminal part of H2AL.2 should be presented, next to the sequence that is replaced.
Reply
The sequence of H2A used to replace the “R-rich motive” of H2A.L.2 is now indicated.
b-c) Words on x-axes cannot be read, please change. Also indicate levels of significance next to the error bars.
Reply
This figure was revised and the requested information is now given.
b) It is stated that the dynamics of H2A.L.2 is lost upon the replacement of the N-terminal part with that of H2A (line 265). Based on Figure 2b it is clear that it is not lost, but reduced.
Reply
“lost” was “replaced by “reduced”.
c) The pictures and the graphs do not seem to match. There seems to be quite a bit of diffuse staining in the H2A.L.2-H2aAnter cells, but I the graphs it is zero. Please explain the discrepancy. Also, the alignment of the pictures is sloppy, same for 3c.
Reply
Sorry for the confusion. Here by diffuse we meant the absence of privileged accumulation in the pericentric regions. In the revised version, the term “diffuse” was explained as the absence of privileged accumulation in the chromocenters.
The problem of figure alignment is probably due to the absence of conversion of the manuscript into pdf format
Line 271 Indicate more specifically which satellite DNA was cloned into the expression vector.
Figure 3a Levels of significance should be indicated.
Reply
In the revised version, we are indicating that two repeat units of the mouse pericentric DNA were cloned in the expression vector. The level of significance is now indicated.
For the collection of sperm, mice were used. It should be confirmed that all experiments have been
Performed in accordance with relevant regulations, and mentioning the committee approving the experiments.
Reply
The following sentence has been now added: For this study the animals were bred in an animal facility with an official and up-to-date agreement according to French and European regulation, and euthanized following a procedure approved by ad hoc committees of the Grenoble Alpes university. All investigators have an official animal-handling authorization obtained after 2 weeks of intensive training and a final formal evaluation. Efforts were made to minimize suffering.
Minor comments.
Articles are lacking at places: line 76 a G26 needle; line 77 at instead of in; line 90 the Qiagen; line 92 the NEB; line 118 for instead of by; line 120 the pellet.
Reply
All was corrected
Reviewer 2 Report
This manuscript is a well written examination of the nuclear localisation of H2A.L.2. The authors have previously identified a domain of H2A.L.2 important for its localisation. Here, the authors show that this protein is peri-centric in spermatozoa. The authors transfect an RFP labelled fusion protein into NH3T3 cells and show that it is localised to peri-centric domains. Expression of a chromo center-disrupting domain ablated this pattern. The authors then show that the N terminal of H2A.L2 and its interaction with RNA is important for this localisation.
The authors look at RNA seq of spermatozoa at different stages of differentiation. They find that levels of major satellite RNA is not changed in H2A.L.2 knockout animals. By disrupting chromocenters, they find that microsatellite is now expressed in these domains and H2A.L.2 is displaced.
This is a rather short but interesting article.
The authors need to explain why the expression of pTVLp64 leads to the expression of microsatellite RNAs at the pericenters yet H2A.L.2 is excluded from these pericenters. If their model was true, there should be more localisation of H2A.L.2 to these centers, which does not occur.
Author Response
The authors need to explain why the expression of pTVLp64 leads to the expression of microsatellite RNAs at the pericenters yet H2A.L.2 is excluded from these pericenters. If their model was true, there should be more localisation of H2A.L.2 to these centers, which does not occur.
Reply
The chimeric protein encoded by pTVLp64 bears a mouse pericentric-sequence specific zinc finger fused to the very strong VP16 transcriptional activator domain. The pericentric targeting of the fusion protein basically destroys the chromocenter organization. This is why H2A.L.2 does not form nuclear foci when cells are transfected with the pTVLp64 plasmid. In other words, there could not “be more localization of H2A.L.2 to these centers” simply because they do not exist.
Reviewer 3 Report
In the manuscript “RNA-guided genomic localization of H2A.L.2 histone variant”, the authors show that a special histone variant in male germ cells, H2A.L.2, specifically localizes to pericentric chromatin by binding RNA. The authors demonstrate that H2A.L.2 has an RNA-binding domain and a propensity to localize to pericentric chromatin and both of these features stabilize the H2A.L.2 interactions with the pericentric chromatin.
The manuscript is clearly written and easy to follow. The images are beautiful, and statistics sound and the authors’ conclusions are consistent with their observations. I do not have any major or minor revisions for this manuscript.
Author Response
Thank you for these positive and encouraging comments
Reviewer 4 Report
The article by Hoghoughi et al “RNA-guided genomic localization of H2A.L.2 histone variant” represents a highly important study of the role of H2A.L.2 histone variant in histone-to-protamine replacement during late spermatogenesis. The authors demonstrated the protein is preferably localized (and is co-localized with protamines) in DAPI-positive pericentromeric regions of spermatids and mature spermatozoa. The authors created a somatic cell model of ectopic expression and again demonstrated heterochromatin-specific localization of H2A.L.2. In another experiment, the putative RNA-binding domain of H2A.L.2 was replaced and even stronger concentration of the protein in the pericentric regions was demonstrated, suggesting the involvement of RNA-binding in the control of H2AL2 tropism. I found the article very interesting and highly recommend it for publication after revising some minor issues indicated below.
Line 74. Please indicate the number of cells/the size of the pellet used for RNA extraction (per 750 ul of Trizol).
Line 115 “0.05mM of … probe” it is more relevant here to indicated the amount of probe (per slide), rather than concentration.
Figure 1 Description of H2A.L.2ko is missing in the figure legend. In proportions of cells with the intranuclear localization graphs – what is “nuclede ” or “nuclecle”?
Line 210 Localization in nucleoli is described, but it remains unclear, how this structure was detected. It is not indicated in the figures.
Author Response
Line 74. Please indicate the number of cells/the size of the pellet used for RNA extraction (per 750 ul of Trizol).
Reply
The amount of RNA obtained is now given.
Line 115 “0.05mM of … probe” it is more relevant here to indicated the amount of probe (per slide), rather than concentration.
Reply
The volume in addition to the concentrations of probes used is now indicated
Figure 1 Description of H2A.L.2ko is missing in the figure legend.
Reply
The reference for H2A.L.2 ko mice is now added in this legend.
In proportions of cells with the intranuclear localization graphs – what is “nuclede ” or “nuclecle”?
Reply
There is no “nuclede ” or “nuclecle” in our text and figures. This is probably because the journal did not convert the submitted manuscript into a pdf format.
Line 210 Localization in nucleoli is described, but it remains unclear, how this structure was detected. It is not indicated in the figures.
Reply
A sentence was added in the legend of Figure 1 to clarify this point
Nuclear regions accumulating RFP-H2A.L.2 in the relatively large nuclear domains (3-4) not stained by DAPI (that contrast with the DAPI-bright chromocenters) were considered as nucleoli.